# Heterogeneity of the Clinical Presentation of the MEN1 LRG_509 c.781C>T (p.Leu261Phe) Variant Within a Three-Generation Family

**DOI:** 10.3390/genes12040512

**Published:** 2021-03-31

**Authors:** Aleksandra Gilis-Januszewska, Anna Bogusławska, Kornelia Hasse-Lazar, Beata Jurecka-Lubieniecka, Barbara Jarząb, Anna Sowa-Staszczak, Marta Opalińska, Magdalena Godlewska, Anna Grochowska, Anna Skalniak, Alicja Hubalewska-Dydejczyk

**Affiliations:** 1Department of Endocrinology, Jagiellonian University Medical College, 31-008 Cracow, Poland; boguslawskaania@gmail.com (A.B.); anna.sowa-staszczak@uj.edu.pl (A.S.-S.); mkal@vp.pl (M.O.); magdalena.godlewska@student.uj.edu.pl (M.G.); grochowskaamd@gmail.com (A.G.); anna.skalniak@uj.edu.pl (A.S.); alahub@cm-uj.krakow.pl (A.H.-D.); 2Department of Nuclear Medicine and Endocrine Oncology, Maria Skłodowska-Curie Institute–Oncology Centre, Gliwice Branch, 44-102 Gliwice, Poland; Kornelia.Hasse-Lazar@io.gliwice.pl (K.H.-L.); Beata.Jurecka-Lubieniecka@io.gliwice.pl (B.J.-L.); Barbara.Jarzab@io.gliwice.pl (B.J.)

**Keywords:** cascade genetic screening, hyperparathyroidism, multiple endocrine neoplasia type 1, pancreatic neuroendocrine tumor, pituitary tumor

## Abstract

Multiple neuroendocrine neoplasia type 1 (MEN1) is a rare genetic disorder with an autosomal dominant inheritance, predisposing carriers to benign and malignant tumors. The phenotype of MEN1 syndrome varies between patients in terms of tumor localization, age of onset, and clinical aggressiveness, even between affected members within the same family. We describe a heterogenic phenotype of the *MEN1* variant c.781C>T (LRG_509t1), which was previously reported only once in a family with isolated hyperparathyroidism. A heterozygous missense variant in exon 4 of the gene was identified in the sequence of the *MEN1* gene, i.e., c.781C>T, leading to the amino acid change p.Leu261Phe in a three-generation family. In the screened family, 5/6 affected members had already developed hyperparathyroidism. In the index patient and two other family members, an aggressive course of pancreatic neuroendocrine tumor (insulinoma and non-functioning neuroendocrine tumors) with dissemination was diagnosed. In the index patient, late diagnosis and slow progression of the disseminated neuroendocrine tumor have been observed (24 years of follow-up). The very rare variant of *MEN1*, LRG_509t1 c.781C>T /p.Leu261Phe (LRG_509p1), diagnosed within a three-generation family has a heterogenic clinical presentation. Further follow-up of the family members should be carried out to confirm the spectrum and exact time of clinical presentation.

## 1. Introduction

Multiple neuroendocrine neoplasia type 1 (MEN1) is a rare, heterogeneous genetic disorder with an autosomal dominant inheritance, predisposing the carrier to benign and malignant tumors. The prevalence of MEN1 is estimated to be approximately 1:30,000 in individuals, with no sex predominance [1]. The penetrance of the disease is very high and increases with age (up to 95% by the age of 50 years). The phenotype of MEN1 syndrome varies between patients in terms of tumor localization, age of onset, and clinical aggressiveness, even between affected members within the same family [2,3,4,5,6,7,8]. The most common endocrinological manifestations are parathyroid adenoma, pancreatic neuroendocrine tumor, and pituitary adenoma. Other characteristic non-endocrine features include cutaneous lesions such as angiofibromas, collagenomas, and café au lait macules [1,9]. Most cases of multiple endocrine neoplasia type 1 occur in a familial setting; however, in 10% of patients, de novo variants have been described and single reports of mosaic *MEN1* alterations have been noted [10,11]. The diagnosis of MEN1 syndrome can be clinically established based on (i) the presence of two tumors typical of MEN1; (ii) the presence of one characteristic MEN1 tumor and one first-degree relative with a confirmed disease-associated *MEN1* variant, or (iii) cascade genetic screening in an asymptomatic carrier [12]. Genetic screening tests should search for sequence variations or large deletions. Genetic screening is recommended starting at the age of 5 years [13,14]. 

The *MEN1* gene, located on chromosome 11q13, encodes the menin protein [15,16]. Menin directly regulates expression of the cyclin-dependent kinase-inhibiting (*CKI*) genes *CDKN1b* (encoding p27) and *CDKN2C* (encoding p18). To date, over 1500 variants of *MEN1* have been described, including frameshift variants, nonsense variants, missense variants, splice site variants, and large deletions. Additionally, patients with these variants should be carefully monitored due to their uncertain clinical significance and heterogeneous presentation. [4,5]. In the present study, we describe a heterogenic phenotype of the *MEN1* variant c.781C>T, previously reported only once in a family with isolated hyperparathyroidism [17].

## 2. Materials and Methods

Extraction of DNA from whole blood of the patients, which was collected on EDTA, was performed using the NucleoSpin^®^ Blood kit (Macherey-Nagel, Dueren, Germany). All coding exons of *MEN1* (LRG_509t1) were amplified using tiTaq DNA polymerase (EurX, Gdansk, Poland) and 100 ng DNA in a 25-µL reaction in an Eppendorf MasterCycler thermocycler. Following electrophoresis (Biometra Horizon 20.25 with a Consort EV243 power pack) on a 1% agarose gel stained with ethidium bromide, the obtained PCR products were visualized by a gel documentation system with a UV transilluminator (UVP GelDoc IT310). After purification of the PCR products through the use of a commercially available kit (NucleoSpin^®^ Gel and PCR Clean up, Macherey-Nagel, Dueren, Germany), the fragments were sequenced using 25 ng of the purified PCR product and the BigDye^®^ Terminator v3.1 Cycle Sequencing Kit (LifeTechnologies, ThermoFisher Scientific, Waltham, MA, USA), purified by ethanol precipitation. These fragments were then read on an ABI3500 sequencer (Applied Biosystems, ThermoFisher Scientific, Waltham, MA, USA). The obtained sequences were aligned to the appropriate reference using SeqScape software (LifeTechnologies, ThermoFisher Scientific, Waltham, MA, USA). Further analyses were performed using the UCSC Genome Browser (http://genome.ucsc.edu/, accessed on 24 July 2020) and associated links on the human GRCh38/hg38 genome assembly. Multiplex ligation-dependent probe amplification (MLPA) analysis (MRC Holland probemix P017) was performed according to the manufacturer’s recommendations in order to rule out large deleterious events in the gene. PROVEAN (v1.1.3) and SIFT (v4.0.3) tools (provean.jcvi.org) and the Bayes classifier-based MutationTaster prediction tool (www.mutationtaster.org) (all accessed on 24 July 2020) were used for prediction of the identified variant’s impact on the protein’s biological function. 

## 3. Results

A large Polish family was identified in which three generations had a *MEN1* variant as confirmed by genetic testing. The index patient, II.3 of the pedigree (Figure 1), was admitted to our Department at the age of 54 and MEN1 diagnosis was based on clinical manifestations (Table 1). Genetic testing was performed in the index patient at the age of 61. Genetic screening revealed a *MEN1* pathogenic variant in other five family members, four of whom had clinical symptoms.

### 3.1. Index Patient

The index patient was a female born in 1953 and was the mother of two unaffected children. She was first admitted in 1996 to the Internal Medicine Department due to symptomatic hypoglycemia, which she had observed for ten months. An insulin-secreting neuroendocrine tumor was suspected based on biochemical as well as clinical data. A computed tomography (CT) scan of the abdomen revealed a pancreatic tumor. The patient underwent surgical pancreatoduodenectomy (Traverso-Longmire procedure). An insulin-secreting neuroendocrine tumor was confirmed by histopathology. Additionally, in 1996, the patient underwent cholecystectomy due to cholecystolithiasis. During that time, hypercalcemia was not found in the patient’s medical records; however, ultrasound examination of the kidneys suggested nephrolithiasis. After surgery, the patient developed diabetes, which was poorly controlled with intensive insulin therapy. In 1999, the patient underwent a total hysterectomy with bilateral salpingo-oophorectomy due to a myoma of the uterus. 

The diagnosis of primary hyperparathyroidism (PHPT) was confirmed in 2007. At this point, the patient had already suffered from nephrolithiasis for several years. Ultrasound examination of the neck revealed a nodular goiter and a single hypoechogenic mass in the topography of the left inferior parathyroid gland. The patient underwent a partial strumectomy and resection of the enlarged parathyroid. Parathyroid adenoma was identified via histopathology. The patient was diagnosed with MEN1 syndrome based on the clinical diagnostic criteria. Pituitary magnetic resonance imaging showed a normal pituitary gland (size 12 × 6 × 14 mm) with no sign of focal lesions. In 2008, during the radiological follow-up, abdominal CT scan revealed focal lesions—possible metastases—located ventrally to the connection of the splenic and the inferior mesenteric vein, in the remaining part of the duodenum, and in the mesenteric lymph nodes. The somatostatin receptor imaging (SRI) with ^99m^Tc result was negative. Repeat surgery was performed, with resection of masses in the mesentery as well as those located near the splenic vessels. Histopathological analysis revealed metastases of the neuroendocrine tumor infiltrating adipose tissue, Ki67 < 1%; 90% of the cells had positive staining for somatostatin receptor (SSTR) type 2, type 4, and type 5, and 50% of the tumor cells showed positive staining for SSTR type 3. Post-operative SRI detected no pathological accumulation of the tracer. 

During a follow-up visit in 2008, biochemical relapse of primary hyperparathyroidism was detected (parathormone (PTH) 163.8 pg/mL, total calcium concentration 2.81 mmol/l) without any visible lesions in ultrasonography or single-photon emission computed tomography/computed tomography (SPECT/CT) with technetium-99 m methoxyisobutylisonitrile (^99m^Tc) Tc-MIBI. In 2011, repeat SPECT/CT showed a pathological accumulation of ^99m^Tc-MIBI in the topography of the left lower parathyroid. In 2011, the patient underwent a total parathyroidectomy, resulting in subsequent iatrogenic long-lasting hypoparathyroidism. 

In 2017, a second relapse of the neuroendocrine tumor was observed; (^68^Ga) Ga-DOTATATE positron emission tomography/computed tomography (PET/CT) revealed the pathological expression of somatostatin receptors in an epigastric lymph node and in the second segment of the liver. In 2018, abdominal CT detected hypodense focal lesions in the liver and enlarged mesenteric lymph nodes (Figure 2a–c). Moreover, mesenteric lymph nodes and a focal lesion in the liver hilum (size 18 × 15 mm) were detected. A non-functional tumor (18 × 10 mm) was detected in the left adrenal gland. Magnetic resonance imaging of the liver revealed lesions in the eighth segment (20 × 30 mm) and the second segment (11 mm). These were suspected to be metastatic lesions. 

Fluorine-18 fluorodeoxyglucose PET/CT (FDG-PET) revealed a metabolically active region in the liver and in the mesogastric region, either in the small intestine or in a lymph node. Subsequently, (^68^Ga) Ga-DOTATATE PET/CT was performed. In the liver, no pathological accumulation of the tracer was found; however, lymph nodes along the lesser curvature of the stomach showed pathological expression of SSTR. At this point, the patient did not consent to any further invasive diagnostic procedures. Physical examination revealed the presence of cutaneous collagenomas and hyperpigmented skin lesions (Figure 2d). 

The patient’s mother is 84 years old (patient I.2). Calcium and PTH concentrations in serum are within the normal range. She suffers from chronic gastritis but is in good general condition. The patient’s mother did not consent to genetic testing. The patient’s father suffered from prostatic hyperplasia and died at the age of 84 due to a pulmonary infection. Genetic testing was not performed. The index patient has two siblings: a brother with MEN1 syndrome (patient II.4 in the pedigree) and a sister (patient II.1) who is in good general condition. Her total calcium and PTH concentrations in serum are within the normal range. Genetic testing has not yet been performed.

### 3.2. Patient II.4

The index patient’s brother was born in 1955 and is the father of five children, two of whom are unaffected. He was first admitted to the Endocrinology Department in 2015 and MEN1 syndrome was confirmed via familial genetic screening. During the patient’s first endocrinological evaluation, hypercalcemia (2.7 mmol/L) and elevated PTH (230 pg/mL) were detected. A nodular goiter and an enlarged right upper parathyroid gland were seen in the thyroid ultrasound examination. The right lobe and the isthmus as well as the parathyroid gland were resected in 2016. Histopathological examination revealed parathyroid adenoma; however, malignancy of the thyroid gland was not detected. Since the surgery, calcium concentration and PTH levels have been within the normal range and the patient has no signs or symptoms of nephrolithiasis or osteoporosis. Multiple MRI examinations performed during follow-up visits (most recently in 2017) did not detect any focal lesions of the pituitary gland or excessive hormone secretion. An abdominal CT from 2016 revealed a 25-mm tumor of the uncinate process of the pancreas. Subsequent SRI confirmed an abnormal accumulation of the tracer in this location. In 2017, the patient did not consent to endoscopic ultrasound nor to surgical resection of the pancreatic lesion. The patient did not attend any follow-up visits until October 2019, when another abdominal CT was performed. Imaging revealed that the pancreatic lesion remained stable; however, in (^68^Ga) Ga-DOTATATE PET/CT, the lesion showed intense accumulation of the tracer (SUVmax 52). Additionally, pathological expression of the somatostatin receptors (SUVmax 3.32) in the marrow cavity of the proximal 1/3 of the femur was discovered. Subsequent CT confirmed the presence of a focal lesion in this location (32 mm), which was suspected to be metastatic. Additionally, suspicious lesions with pathological expression of the somatostatin receptors were discovered in the femur and tibia. The patient did not consent to any further diagnostic procedures.

### 3.3. Patient III.3

A daughter of the index patient’s brother was born in 1979 with congenital hypoacusia and was identified as carrier of a *MEN1* during familial screening. In 2014, she underwent a urological procedure due to nephrolithiasis in the right kidney and was subsequently diagnosed with primary hyperparathyroidism. A subtotal parathyroidectomy was performed in 2014. In 2018, the patient underwent a partial thyroidectomy due to presence of a nodular goiter. During a follow-up visit in 2019, CT showed a 5-mm hypodense focal lesion in the corpus of the pancreas. A similar 4-mm lesion was found in the uncinate process. Imaging follow-up studies of the lesions and careful clinical and biochemical assessments have been scheduled. Additionally, physical examination revealed multiple cutaneous angiofibromas on the face and hyperpigmented skin lesions (Figure 3.) This patient has two daughters, both of whom are negative for *MEN1*. 

### 3.4. Patient III.5

A son of the index patient’s brother, born in 1980, was diagnosed with MEN1 syndrome via cascade genetic screening. Since 2014, he had been suffering from nephrolithiasis and underwent percutaneous nephrolithotomy. Hypercalcemia in the course of primary hyperparathyroidism was confirmed and the patient underwent parathyroidectomy with resection of the upper right, lower right, and lower left parathyroid glands in April 2015, which subsequently resulted in iatrogenic hypoparathyroidism. Histopathological analysis confirmed the presence of parathyroid nodular hyperplasia. In 2014, bilateral adrenal tumors were detected in abdominal CT (24 mm in the left adrenal gland and 5 mm in the right, both characterized by radiological features of an adenoma), with no excessive hormonal function. Additionally, CT confirmed the presence of a pancreatic cyst, which was left in place for radiological follow-up. In 2015, MRI revealed a pituitary microadenoma, clinically and biochemically silent. During a follow-up visit in 2016, abdominal CT revealed a round 8-mm tumor with calcifications in the corpus of the pancreas (Figure 4a). 

In November 2017, (^68^Ga) Ga-DOTATATE PET/CT was performed. It revealed multiple focal accumulations of the tracer in the pancreas, with the most prominent one observed within the pancreatic corpus (SUVmax 22.8), located in the topography of a previously described calcified tumor, and a probable metastasis in the left lung (SUVmax 6.54). In January 2018, due to dissemination of the neuroendocrine tumor, treatment with Lanreotide autogel at 120 mg s.c./4 weeks was initiated with good patient tolerance. In December 2018, an endoscopic ultrasound was performed. Histopathological examination of the samples revealed the presence of a probable neuroendocrine tumor (synaptophysin+, chromogranin+). In January 2020, follow-up CT revealed a partial regression of the pancreatic lesions, while the pancreatic neuroendocrine tumors and metastatic lung lesion were stable with no further progression (Figure 4b,c). Careful follow-up has been scheduled. Non-endocrine manifestations were seen as multiple cutaneous collagenomas. 

This patient has three sons. Two are unaffected, while one is an asymptomatic carrier of the familial *MEN1* variant, which was diagnosed at the age of 7 years. 

### 3.5. Patient III.9

A daughter of the index patient’s brother, born in 1991, was diagnosed at the age of 23 with MEN1 syndrome via cascade genetic screening. At the time of diagnosis, asymptomatic hypercalcemia was noted and primary hyperparathyroidism was confirmed. She underwent total resection of three parathyroid glands in 2015. During a follow-up visit in 2020, abdominal CT revealed two pancreatic lesions (17 mm and 15 mm). Subsequently, endoscopic ultrasound-guided fine-needle aspiration biopsy of the pancreas was performed. Histopathological examination revealed the presence of a neuroendocrine tumor (CD56-, Ki67 1%, synaptophysin+, chromogranin+). Radiological follow-up has been scheduled. In addition, the patient has multiple cutaneous collagenomas. 

### 3.6. Genetic Results

A heterozygous missense variant in exon 4 was identified in the sequence of the *MEN1* gene (LRG_509), i.e., c.781C>T, leading to the amino acid change p.Leu261Phe at the protein level. This variant has been registered in the NCBI dbSNP with the identifier rs878855198. Based on criteria provided by a single submission to NCBI ClinVar, this variant (ID 241826) is classified as having uncertain clinical significance. In the LOVD database (ID MEN1_000346), this alteration has no assigned clinical classification. In the UMD-MEN1 database, there are three entries for this variant (UMD_ID 557, 558, and 595), two of which relate to samples derived from members of one family. In all three cases, the variation has been classified as “causal”. The variant is located in a phylogenetically conserved region, and leucine at position 261 of the protein is, itself, highly conserved (Figure 5). Predictions performed through the use of bioinformatics tools generate consistent results. PROVEAN classifies this variant as “deleterious” (score −2.86), SIFT classifies this as “damaging” (score 0.011), and MutationTaster classifies this as “disease-causing” (with a probability of >0.999). 

## 4. Discussion

Our study describes the diverse outcomes (aggressive and benign) of a very rare *MEN1* variant within the same three-generation family. In this family, 5/6 affected members have already developed hyperparathyroidism, with the earliest age at presentation being 23 years, while the oldest presented at the age of 59 years. The youngest family member was confirmed to carry the familial variant at the age of 7. This family member is now 11 years of age and has been asymptomatic up until this point. Three of the family members have shown an aggressive course of the disease with dissemination of pancreatic neuroendocrine tumors; however, the index patient was diagnosed relatively late and her disease progression is categorized as very slow (24 years up until now). In only one affected family member, pituitary microadenoma was detected at the time of diagnosis (at age 34) along with other manifestations of the syndrome (hyperparathyroidism, non-functioning pancreatic tumor). To date, the MEN1 LRG_509 c.781C>T (p.Leu261Phe) variant has been described in one family presenting with a mild phenotype, with isolated hyperparathyroidism reported as a gene variant with uncertain significance [4,5]. In the literature, there are several reports of MEN1 alterations related only to isolated hyperparathyroidism, while the same variant in other families results in different phenotypes with other MEN1-associated tumors [4,5]. Despite the large number of identified variants in *MEN1*, genotype–phenotype correlation has not been estimated due to the heterogeneity of the disease. Previous studies of unrelated patients with the same variant demonstrated the variable clinical expression of MEN1 characteristic tumors. Even case reports of monozygotic twins have reported different phenotypes of MEN1 syndrome [18,19]. It is hypothesized that environmental factors could have an influence on epigenetic effects and, therefore, lead to various clinical outcomes. Patients diagnosed with MEN1 syndrome should be treated and managed according to current guidelines [12]. Screening of the presented family was scheduled at the time of diagnosis of the index patient; however, these examinations were performed in most of the family members in adulthood, after a long delay. Screening in childhood, as suggested by guidelines, allowed for identification of an asymptomatic carrier (patient IV.4) and for scheduling of the appropriate follow-up. MEN1 patients have a decreased life expectancy and worse clinical outcomes using currently available treatment when compared to sporadic cases with corresponding tumors. MEN1-associated tumors are often multifocal and more aggressive, with distant metastases, and are, therefore, resistant to treatment [20,21]. 

The earliest and most common MEN1-related feature is primary hyperparathyroidism due to parathyroid hyperplasia or parathyroid adenoma. Up to the age of 50 years, hyperparathyroidism shows almost 100% penetrance. Conversely, among all patients diagnosed with PHPT, *MEN1* variants are observed in 1–18% of patients [22]. Hyperparathyroidism associated with MEN1 is more aggressive (e.g., a greater decline in bone mineral density as well as urolithiasis at a young age) [23]. In addition, multiple involvement of the gland and higher recurrence rate have been observed in comparison to sporadic cases (55% vs. 4–16%) [24]. Four affected members in our study presented with PHPT as a first manifestation and developed urolithiasis at a young age. In the index patient, the first manifestation was the presence of an insulinoma, while cholelithiasis and urolithiasis preceded the diagnosis of hypercalcemia. In the presented family, urolithiasis and cholelithiasis without hypercalcemia preceded the diagnosis of hyperparathyroidism by several years. Therefore, in the case of MEN1-related tumors, urolithiasis, or cholelithiasis without hypercalcemia, should raise a suspicion of PHPT and lead to further investigation. Pancreatic neuroendocrine tumor is the second most common MEN1-related tumor and is present in 30–80% of MEN1 patients; however, post-mortem studies showed a higher prevalence of 80–100%. Ten percent of all pancreatic neuroendocrine tumors (pNETs) occur in the course of MEN1 syndrome [25]. The most common pNETs are non-functioning tumors (20–55% of MEN1 cases) [26]. Insulinomas occur in approximately 20–30% of cases, with a metastatic process being observed in 4–14% of MEN1 patients [12,27,28]. The diagnosis of an insulinoma can be challenging. In some cases, especially if no visible pancreatic lesion is observed, advanced imaging techniques are required to detect the insulinoma [29]. In the index patient, the presence of an insulinoma was confirmed at the age of 43, while dissemination was detected at the age of 54. In addition, patients II.4 and III.5 presented with disseminated non-functional NETs. In patients III.3 and III.9, non-functional pancreatic NETs remain monitored at follow-up visits. The future treatment options depend on the course of the disease. Surgery, peptide receptor radionuclide therapy (PPRT), and pharmacotherapy are taken into consideration.

Pituitary tumors associated with MEN1 occur mostly in the fourth decade of life [20,21]. Prolactinoma is the most frequent pituitary manifestation; however, non-functioning pituitary adenomas were predominantly detected by screening and were stable lesions [30]. A non-functioning pituitary microadenoma in patient III.5 was confirmed at the time of MEN1 diagnosis. Non-functioning adrenocortical tumors were present in 2/6 of the affected patients. Characteristic MEN1-related skin lesions (angiofibromas, collagenomas, hyperpigmentation) were present in all of the screened family members. Nodular goiter was present in 4/6 of the family members. Careful skin examination and diagnostics for other tumors should be performed in all suspected MEN1 patients. The association between thyroid disease and MEN1 remains controversial. In the earlier studies, nodular goiters were linked to MEN1, while the more recent ones show that the rate of concurrence of a thyroid incidentaloma is comparable in MEN1-positive and -negative patients [31]. Patient III.3 presented with hypoacusia. Thus far, there are no published reports of MEN1 and hypoacusia; however, various syndromic diseases associated with hearing loss have been reported [32,33]. 

A limitation of our study is the fact that family members of the index patient were not followed-up prospectively but were diagnosed at a certain point in time, long after the index patient was diagnosed. Therefore, the exact onset of the MEN1 manifestations could not be established. The delay in the cascade genetic screening resulted from a delayed transfer of information about the genetic background of the disease between the family members. The probable psychological barriers which hinder the sharing of information about genetic diseases need further investigation and support [34]. 

## 5. Conclusions

The very rare variant of *MEN1*, LRG_509t1 c.781C>T /p.Leu261Phe (LRG_509p1), diagnosed within a three-generation family, has a heterogenic clinical presentation. This included the presence of previously unreported neuroendocrine tumors in selected family members. Further follow-up of the family members should be performed to confirm the spectrum and exact time of clinical presentation of this alteration. 

## Figures and Tables

**Figure 1 genes-12-00512-f001:**
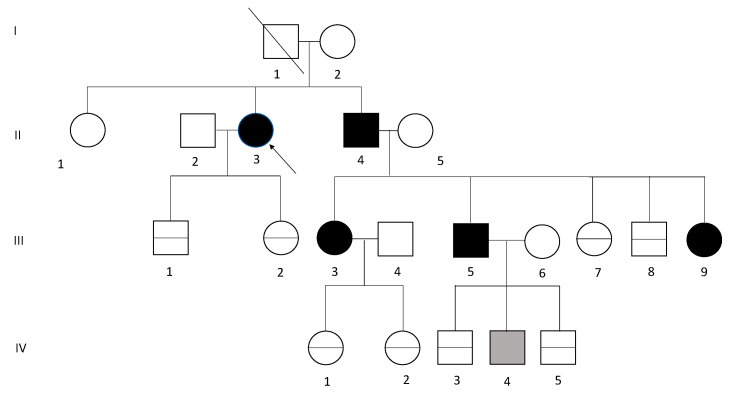
Pedigree of a family with multiple neuroendocrine neoplasia type 1 (MEN1) syndrome. Generations available for the study are indicated by Roman numerals I–IV. Black symbols: affected subject. Grey symbol: *MEN1* variant carrier. Squares indicate male; circles indicate female; the arrow indicates the index patient. The horizontal line in circles and squares indicates a negative *MEN1* variant gene test, while a diagonal line in the square indicates that the patient was not alive at the time of the study.

**Figure 2 genes-12-00512-f002:**
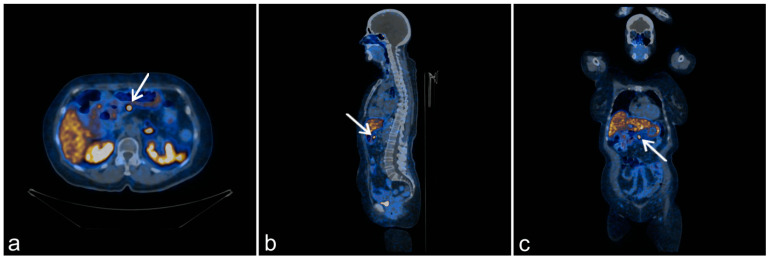
Clinical and radiological presentation of the index patient. Transverse (**a**), axial (**b**), and sagittal (**c**) fused (^68^Ga) Ga-DOTATATE positron emission tomography (PET)/computed tomography (CT) images of a pancreatic insulinoma with metastases (arrows), cutaneous manifestations of MEN1 syndrome (**d**) with collagenoma, and hyperpigmented skin lesions (arrows).

**Figure 3 genes-12-00512-f003:**
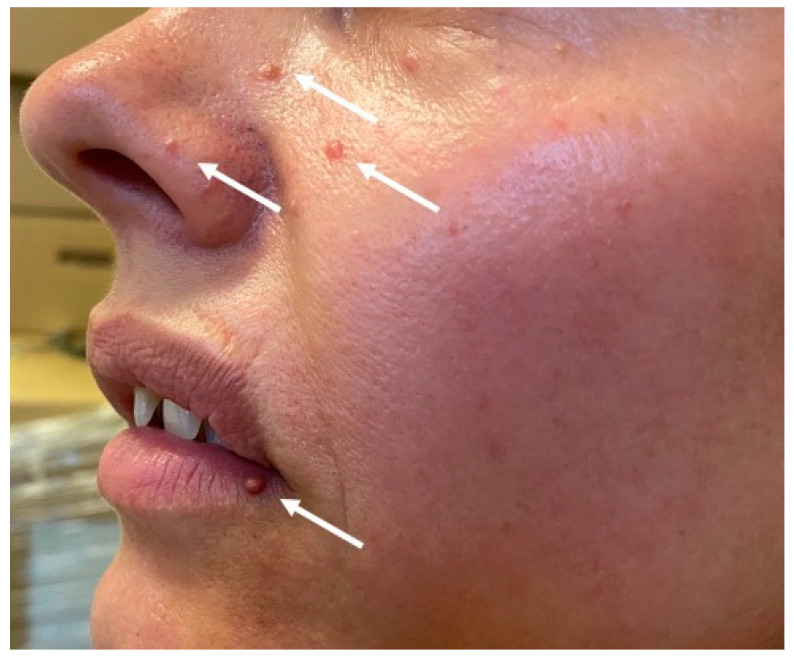
Cutaneous angiofibromas of patient III.3 in the pedigree (arrows).

**Figure 4 genes-12-00512-f004:**
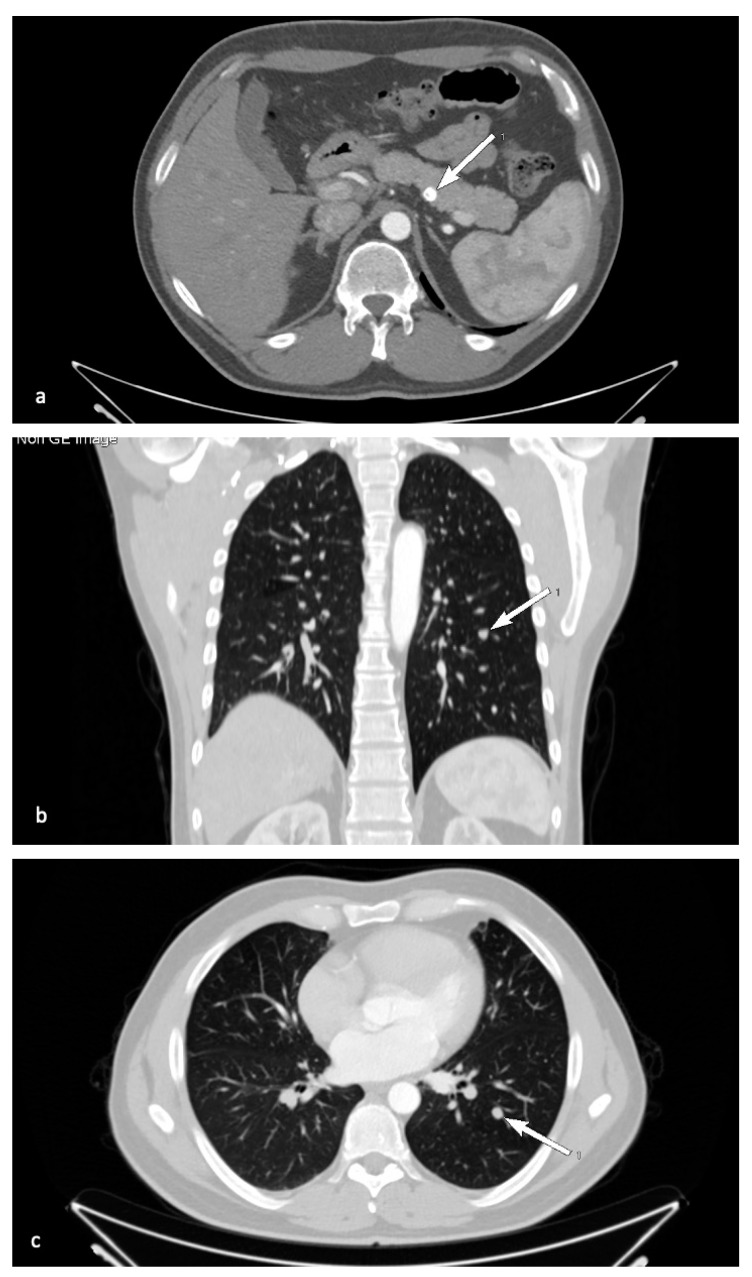
Radiological presentation of patient III.4 in the pedigree. Transverse CT image of pancreas with calcifications (**a**) and lung metastasis in axial (**b**) and transverse (**c**) CT (arrows).

**Figure 5 genes-12-00512-f005:**
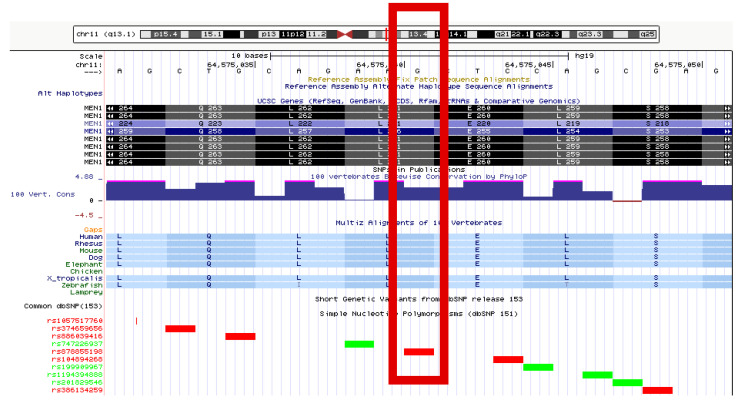
**The** c.781C>T (p.Leu261Phe) variant visualized in the UCSC Genome Browser (genome.ucsc.edu (accessed on 24 July 2020)), indicated with a red box. Phylogenetically conserved regions according to PhyloP and the alignment across different vertebrate species are shown.

**Table 1 genes-12-00512-t001:** Clinical features of the proband and relatives.

Patients	Sex	Age at Diagnosis	Primary Hyperparathyroidism	Kidney Stones	Pituitary Adenoma	Pancreatic Tumor	Adrenal Tumor	Other Manifestations
II.3	F	54	Yes	Yes	No	Yes	Yes	Cutaneous collagenomas, myoma of the uterus
II.4	M	60	Yes	Yes	No	Yes	Yes	Nodular goiter
III.3	F	35	Yes	Yes	No	Yes	No	Cutaneous angiofibromas, nodular goiter, hypoacusia
III.5	M	34	Yes	Yes	Yes	Yes	Yes	Cutaneous collagenomas, nodular goiter
III.9	F	23	Yes	No	No	Yes	No	Cutaneous collagenomas

## Data Availability

Not acceptable.

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
