# Peer review of "Heterogeneity of the Clinical Presentation of the MEN1 LRG_509 c.781C>T (p.Leu261Phe) Variant Within a Three-Generation Family"

_genes, 2021, doi:10.3390/genes12040512_

Round 1
Reviewer 1 Report
Thank you for the opportunity to review the manuscript “Heterogeneity of the clinical presentation of the MEN1 2 LRG_509t1 c.781C>T (p.Leu261Phe) variant within a three-generation family.” by Aleksandra Gilis-Januszewska et al. MEN1 is a rare and heterogeneous disorder. The authors report a three-generation family with a heterogenic phenotype of the MEN1 variant c.[781C>T](LRG_509t1). This variant was previously reported in a family with isolated hyperparathyroidism. Therefore, the manuscript reveals a novel genetic background of MEN1. I find the report very interesting and valuable.
However, there are some issues that need to be improved.
- There is lack of comment of further possible prognosis and possible methods of treatment (paragraph 3.3-3.5, Discussion). In my opinion it should be part of the manuscript even if these are not certain.
- Line 55: “a part of which”- a part of them? The style make it difficult to understand the sentence with an important message.
- Line 84: “enrolled into the study at the age of 54”- isn’t it the age of first clinical diagnosis of MEN1? The research regards mainly the genetic diagnosis and comparing it with the clinical course so the enrolment date is rather the date of patient’s informed consent for the genetic testing.
- There is no information when the genetic tests were done within the family (Materials and Methods or Genetic results). In the Genetic results paragraph- there should be information that the variant was diagnosed in all clinically affected members of the family, if I understand it correctly. There is information in the separate paragraphs regarding the other members tested for the diagnosed variant but I cannot see the clear picture was the genetic testing done in all the members?
- Table 1: Is multiple goitre an additional manifestation of MEN1 or is it just a coincidence? There are 3 cases of nodular goitre in the Table 1 and 4 in the lines 323-324.
- Table 1: patient III.3: Shouldn't it be hypoacusia? Is it inherited or congenital (like in the line 187)? Again, is it a manifestation on MEN1?
- Line 151-152: “She has cutaneous collagenomas and hyperpigmented skin lesions (Figure 2B).” The sentence is not connected to the previous one. Maybe it would be better to write: Additionally, she has... It should be Fig 2d not 2B.
I cannot see mentioning Fig. 2b,c in the text.
Line 229: Fig. 4b,c?
- Overall, I find the description of radiological findings partially too detailed (paragraphs 3.1, 3.2, 3.4). It would be easier to follow if they are more concise.
- Line 339 Courtesy of – is it related to the presented results or images?
Reviewer 2 Report
This report examines the heterogenic clinical presentation of a very rare variant of MEN1 [LRG_509t1 c.781C>T (p.Leu261Phe)] within a three-generation family. In general, the topic is well presented and explained and the follow-up of the patients is clear. However, some minor changes should be done.
Minor concerns:
- English language should be improved along the manuscript.
- Bibliography and introduction sections should be improved including last articles published about MEN1 variants (e.g., Romanet et al., JCEM, 2019; Hu et al., Horm Metab Res, 2020).
- Figures 2B-C are not mentioned in the manuscript.
- Line 152: is Figure 2D instead of Figure 2B.
